# Choroid structure analysis following initiation of hemodialysis by using swept-source optical coherence tomography in patients with and without diabetes

Hideyuki Nakano[1]*, Hiruma Hasebe[1], Kenji Murakami[2], Hiroyuki Cho[1,3], Daisuke Kondo[4], Noriaki Iino[5], Takeo Fukuchi[1]

1 Department of Ophthalmology, Niigata University Graduate School of Medical and Dental Science, Niigata, Japan, 2 Department of Ophthalmology, Niigata City General Hospital, Niigata, Japan, 3 Department of Ophthalmology, Uonuma Kikan Hospital, Minamiuonuma, Japan, 4 Department of Nephrology, Niigata City General Hospital, Niigata, Japan, 5 Department of Nephrology, Uonuma Kikan Hospital, Minamiuonuma, Japan

* hyrn1070@med.niigata-u.ac.jp

**Data Availability Statement:** All relevant data are within the manuscript and its Supporting Information files.

## Abstract

We aimed to evaluate choroid structural changes using swept-source optical coherence tomography (SS-OCT) following hemodialysis initiation in diabetic and nondiabetic patients with end-stage kidney disease (ESKD). In this multicenter, prospective, cross-sectional study, diabetic (DM group; 30 eyes; 16 patients) and nondiabetic patients (NDM group; 30 eyes; 15 patients) with ESKD were evaluated after hemodialysis initiation. SS-OCT findings were analyzed using a manual delineation technique and binarization method before the first and last hemodialysis sessions, conducted approximately 2 weeks apart. Subfoveal choroidal thickness changes and mean large choroidal vessel layer thickness were significantly greater in the DM group (−13.3% ± 2.5% and −14.5% ± 5.2%, respectively) than the NDM group (−9.5% ± 3.1% and −9.2% ± 3.4%, respectively; p = 0.049 and p = 0.02, respectively). Binarized SS-OCT analysis revealed that the mean subfoveal choroidal area was significantly larger in the DM group (−21.9% ± 6.5%) than the NDM group (−17.2% ± 5.9%; p = 0.032). The change ratio in mean luminal area values was significantly greater in the DM group (−27.7% ± 8.7%) than the NDM group (−17.7% ± 5.8%; p = 0.007). The DM group exhibited substantial changes in the choroidal layer, possibly reflecting choroidal vascular disorders caused by diabetes.

## Introduction

Changes in the retinal blood vessel bed have been implicated in the pathophysiology of diabetic retinopathy (DR) [1]. Such changes affect the choroid, the primary source of oxygen and nutrients to the outer retina and the only source of blood flow to the avascular macula [2].

**Funding:** The authors received no specific funding for this work.

**Competing interests:** The authors have declared that no competing interests exist.

Vascular disorders caused by diabetic choroidopathy have been implicated in unexplained vision loss in patients with or without retinal vascular disorders [3].

Patients with end-stage kidney disease (ESKD) experience fluid retention. Hemodialysis optimizes the composition and volume of body fluid, decreases body weight and blood pressure, and increases colloid and interstitial osmotic pressures [4]. These changes may affect ocular parameters such as intraocular pressure, central corneal thickness, central retinal thickness, retinal nerve fiber layer thickness, and choroidal thickness [5]. Numerous patients with diabetic nephropathy also have DR [6], choroidopathy, capillary dropout, focal scarring, and leukostasis [7, 8].

Previous studies primarily evaluated choroidal thickness in patients undergoing maintenance dialysis; however, to date, no study has investigated changes in choroidal thickness following the initiation of hemodialysis. In a recent study, Shin YU et al. reported changes in the choroidal vascularity index (CVI) measured using SS-OCT before and after single hemodialysis in patients undergoing maintenance hemodialysis [9], but there is no study that compared a case immediately before "new" initiation of hemodialysis and two weeks after initiation.

Most reports have indicated that choroidal thickness is reduced in patients with diabetes [10] and in those with DR [11] and that the thickness further decreases as DR progresses [12, 13]. In some cases, reduced signal intensity in the retinal pigment epithelium (RPE) prevents accurate identification of the area between the borders of the choroid and sclera [14]. Swept-source optical coherence tomography (SS-OCT) has the potential to resolve these limitations using different wavelength-tunable lasers, a dual-balanced photodetector, and a long wavelength capable of yielding clear images [15].

The choroid includes blood vessels, connective tissue, melanocytes, nerves, and extracellular fluid, and consists of three main vascular layers: choriocapillaris, Satteler's layer, and Haller's layer. Therefore, measurement of total choroidal thickness alone is insufficient for the evaluation of choroidal structural changes. Recently, attempts have been made to visualize the morphologic features of the choroidal vasculature using OCT [16–18].

No previous study has investigated changes in stromal and luminal areas using SS-OCT images with black and white conversion, as well as changes in thickness of the choriocapillaris and deep choroidal vessels by using delineation before and after "new" initiation of hemodialysis. In this study, we compared choroidal structural changes and systemic parameters in patients with ESKD who either did or did not have diabetes before and after the initiation of hemodialysis. Specifically, we performed choroidal layer analysis and binarization of choroidal OCT images using SS-OCT.

## Materials and methods

### Study design

We conducted a cross-sectional, multicenter study at Niigata City General Hospital and Uonuma Kikan Hospital between September 2016 and October 2017. The study protocol was approved by the institutional review boards of Niigata City General Hospital, Uonuma Kikan Hospital, and Niigata University Graduate School of Medical and Dental Science (Approval number 16–036). This study was conducted in accordance with the Declaration of Helsinki. Informed consent was obtained from all subjects.

### Study subjects

We included 31 patients with ESKD who recently underwent hemodialysis at Niigata City General Hospital and Uonuma Kikan Hospital. We included patients showing best-corrected visual acuity > 20/200 and eyes with an axial length (AL) of 22.1–25.9 mm. Patients showing

anterior or posterior segment disease that hindered accurate examinations and ocular surgery and/or retinal laser treatments within 3 months prior to the study and had a history of retinal vein occlusion, glaucoma, AMD, or uvetis were excluded from this study.

Hemodialysis was initiated in all subjects; three or four sessions were conducted per week, approximately six times during their 2-week hospitalization, with strict control of fluid composition and blood pressure (see S1 Fig). The study compared data obtained before the first and following the last dialysis session. All patients were assessed prior to their first dialysis session.

Patients with ESKD were assigned to two groups: those with (DM) and those without (NDM) diabetic nephropathy (16 and 15 patients, respectively). Causes of ESKD in the NDM group included hypertensive nephrosclerosis (n = 8), IgA nephropathy (n = 2), multiple myeloma (n = 1), and chronic failure of unknown etiology other than DM (n = 4). DR severity in the DM group was classified as follows: no DR (NDR; n = 12 eyes), non-proliferative DR (NPDR; n = 13 eyes), and proliferative DR (PDR; n = 6 eyes). All PDR patients had already completed pan-retinal photocoagulation (PRP) laser treatment.

## Clinical study protocol

The following parameters were measured 15 min before the initiation of a hemodialysis session: blood pressure (systolic/diastolic/mean), heart rate, body weight, serum osmolarity, total protein, albumin, blood urea nitrogen, creatinine, electrolytes, and plasma colloid osmotic pressure. Based on the report by Ishibazawa et al. [19], ophthalmologic examinations performed 30 min before the start of dialysis included measurements of visual acuity, intraocular pressure (IOP), anterior chamber depth (ACD), AL, anterior segment and fundus examination, central retinal thickness, and choroidal thickness. Ophthalmic examinations were conducted before initiation of first single hemodialysis, and before the final single hemodialysis session prior to being discharged (S1 Fig).

IOP was measured using Goldmann applanation tonometry. ACD and AL were measured using an OA-2000 (Nidek Co, Ltd, Gamagori, Japan) and an IOL Master 500 (Carl Zeiss Meditec, Dublin CA, United States) at Niigata City General Hospital and Uonuma Kikan Hospital, respectively. According to the Early Treatments Diabetic Retinopathy Study (ETDRS), DR staging was performed based on the results of fundus examination. Fluorescein angiography was conducted, if necessary.

To prevent diurnal variations, we used the results of only the first hemodialysis session of the day (9 AM–1 PM). Additionally, a recent study reported that light-evoked choroidal expansion is lower in diabetic eyes than in healthy eyes [20]. To prevent this effect, we performed the SS-OCT scan in the same slightly darkened room for all subjects.

## Swept-source optical coherence tomography imaging

A three-dimensional horizontal volume (12 mm, 9 mm, 512,256 resolution) macula scan was performed by experienced examiners for all patients by using a swept-source OCT (DRI OCT Triton®, Topcon, Tokyo, Japan). The central macula thickness and subfoveal choroidal thickness were automatically measured by the software of the SS-OCT device (Topcon Fastmap, Topcon Medical System, Paramus, NJ). User-independent thickness maps represented the average of all points in the inner circle (radius 1 mm) of the center of the nine sectors, which was defined by the ETDRS grid. A follow-up function, installed in the SS-OCT, enabled subsequent images to be automatically scanned in the same location.

## Choroidal layer analysis and binarization of choroidal OCT images

Analysis of the choroidal layer was manually performed based on the procedure described by Branchini et al. [16] (Fig 1). Briefly, the cut-off size was set at 100 μm for large choroidal vessels in horizontal/vertical OCT images, which included the fovea. The caliper function installed in the SS-OCT enabled selection of large choroidal vessels with a horizontal measurement of 100 μm or those located close to the central fovea. Choroid thickness was measured using three parameters. The subfoveal choroidal thickness (SCT) was the measurement from the hyper-refractive line that reflected the Bruch's membrane beneath the RPE to the inner surface of the sclera. The large choroidal vessel layer thickness (LCVLT) was defined as being from a horizontal line drawn perpendicular to the deepest point of the large choroidal vessel intersecting with the SCT measurement line to the inner surface of the sclera. Choriocapillaris-medium choroidal vessel layer thickness was obtained by subtracting LCVLT from SCT.

Binarization of OCT images was performed using the ImageJ software (v. 1.47, NIH, Bethesda, Maryland, USA; available at http://imagej.nih.gov/ij/) as per the method described by Sonoda et al. [17, 18] (Fig 2). Briefly, the examined area was 7.5-mm wide with margins of 3.75 mm nasal/superior and 3.75 mm temporal/inferior to the fovea. The examined area vertically extended from the RPE to the inner surface of the sclera, and the borders were selected using the ImageJ ROI manager. The border line is defined using the ImageJ ROI manager, limiting the examined area vertically from the RPE to the inner surface of the sclera. Three

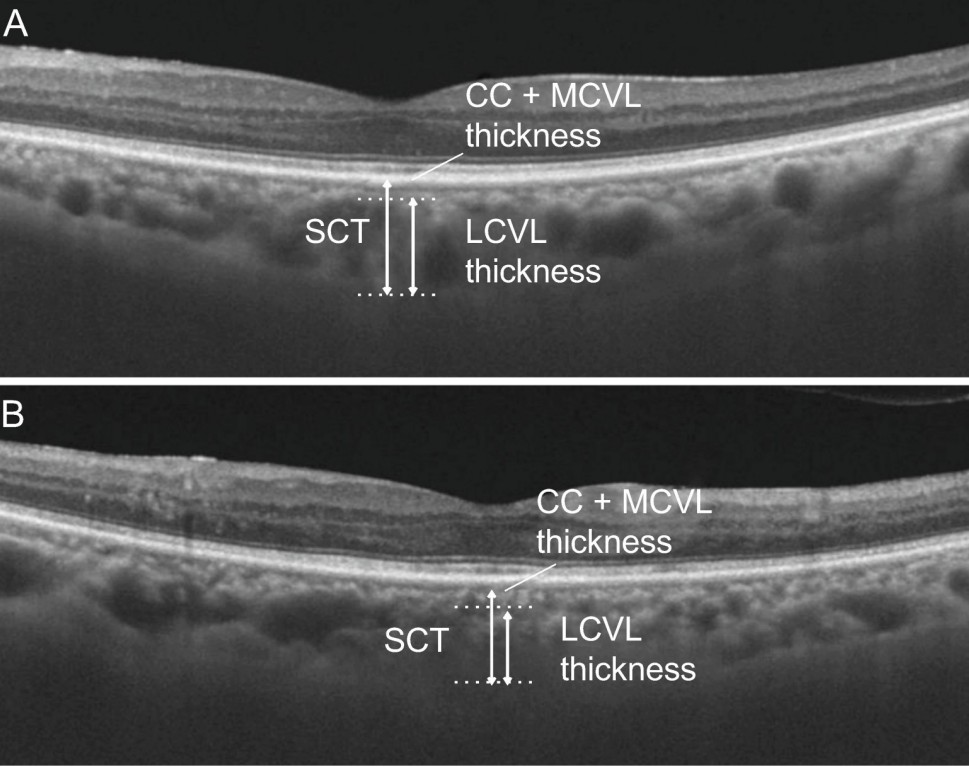

**Fig 1. Swept-source optical coherence tomography images of the choroidal layer.** Horizontal scan of 58-year-old man with diabetic nephropathy is shown. Before the initiation of hemodialysis (A), the subfoveal choroidal thickness (SCT), the large choroidal vessel layer thickness (LCVLT), and the choriocapillaris-medium choroidal vessel layer thickness (CC+MCVLT) were 367, 320, and 47 μm, respectively. After the initiation of hemodialysis (B), SCT, LCVLT, and CC+MCVLT were 335, 297, and 43 μm, respectively.

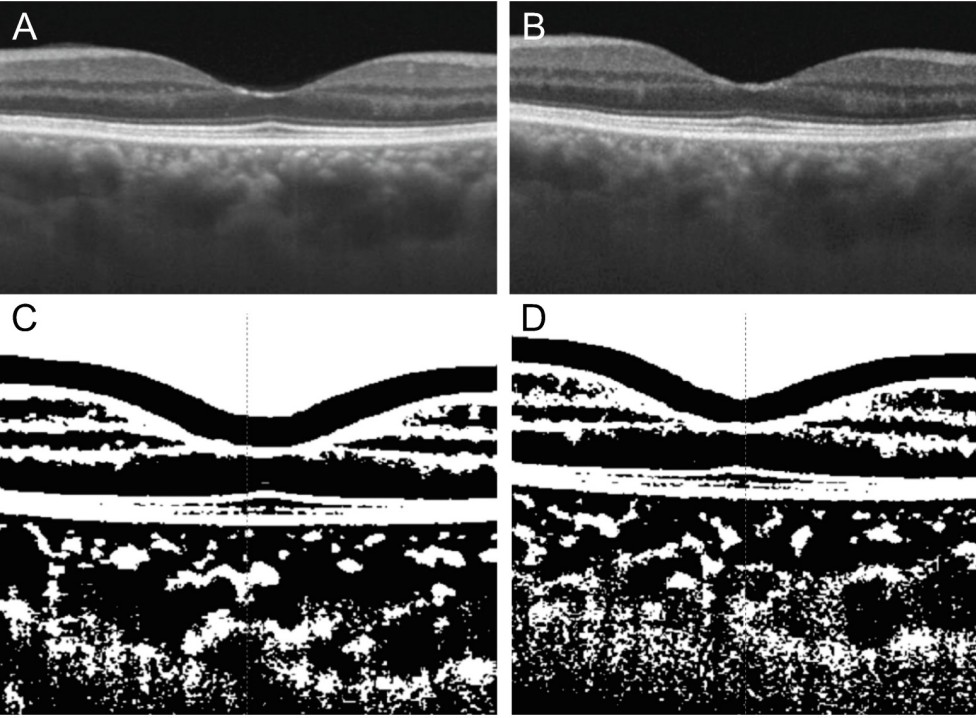

**Fig 2. Binarization of choroidal swept-source optical coherence tomography images.** Horizontal scan of a 64-year-old man with diabetic nephropathy is shown. Before the initiation of hemodialysis (A), using the 7-mm wide binarized SS-OCT image, the choroidal, luminal, and stromal areas were calculated to be 1.753, 1.386, and 0.367 mm$^2$, respectively (C). After the initiation of hemodialysis (B), the choroidal, luminal, and stromal areas decreased to 1.431, 1.122, and 0.309 mm$^2$, respectively (D).

choroidal vessels with lumens of ≥100 μm were selected using the oval selection tool, and the average reflectivity of these luminal areas was determined. Using the oval selection tool to select three choroidal vessels having lumen thickness of 100 μm or more, the average reflectivity was set at the minimum value to minimize noise. Binarization of the choroidal images was performed using Niblack's method. The images were converted to 8-bit images and adjusted by Niblack's Auto Local Threshold. The images were converted to 8-bit images, and adjusted using Niblack Auto Local threshold. The binarized image was converted again into red, green, and blue images, and the luminal area was determined using the threshold tool. The luminal area (LA) and the stromal area (SA) were calculated after adding the distance to each pixel. Light pixels were defined as SA and dark ones as LA. The choroidal layer analysis and binarization of the choroidal images were performed in horizontal and vertical sections, and their values represented the average of all points.

## Statistical analysis

All values are expressed as means ± SD. All values are shown in terms of mean±SD. Systemic hemodynamic parameters and ocular parameters were compared before and after the initiation of hemodialysis using the Wilcoxon signed-rank test, and the two groups were compared using the Mann–Whitney *U*-test. Wilcoxon signed-rank test was used to compare the systemic hemodynamic parameters and ocular parameters before new hemodialysis initiation and two weeks after initiation, while the Mann-Whitney U-test was used to compare the diabetes

group with the non-diabetes group. A generalized linear mixed model was used to compare choroidal thickness and area. Plasma colloid osmotic pressure was calculated as follows: (5.5 × concentration of plasma albumin) + (1.4 × concentration of plasma globulin). The plasma colloid osmotic pressure was calculated as follows: 5.5 × concentration of plasma albumin) + (1.4 × concentration of plasma globulin). The ocular perfusion pressure was shown in terms of mmHg, and it was calculated using the following formula: 2/3 × mean blood pressure —ocular pressure. The choroidal change ratio was calculated as follows: (mean thickness after hemodialysis – mean thickness before hemodialysis)/mean thickness before hemodialysis. All statistical analyses were performed using SPSS version 25 (IBM Corp., Armonk, NY, USA), with $p < 0.05$ being considered significant.

## Results

### Subject demographics and clinical characteristics

The patient characteristics in each group are shown in Table 1. In the DM group, two eyes were excluded owing to glaucoma. In that group, six eyes with PDR received PRP treatment, and six eyes had no DR. The NPDR group included 16 eyes that did not receive PDR treatment.

### Changes in systemic parameters before and after initiation of hemodialysis

After the initiation of hemodialysis, body weight decreased from 70.1 ± 13.9 kg to 66.5 ± 13.5 kg and from 60.0 ± 11.4 kg to 57.8 ± 10.5 kg in the DM and NDM groups, respectively. Systolic BP decreased from 155.6 ± 16.0 mmHg to 137.1 ± 16.8 mmHg (p = 0.04) in the DM group. Mean BP decreased from 102.9 ± 12.9 mmHg to 94.1 ± 12.4 mmHg (p = 0.004) and from 106.8 ± 13.9 mmHg to 94.5 ± 7.3 mmHg (p = 0.003) in the DM and NDM groups, respectively. Serum osmolarity decreased from 307.4 ± 7.5 mOsm/L to 299.4 ± 7.7 mOsm/L (p = 0.024) and from 314.6 ± 7.6 mOsm/L to 302.3 ± 5.9 mOsm/L (p = 0.006) in the DM and NDM groups, respectively. Plasma colloid osmotic pressure increased from 19.9 ± 3.0 mmHg to 21.4 ± 2.8 mmHg (p = 0.03) in the DM group. Changes in patients' systemic values are provided in S1 Table, S1 Fig.

**Table 1. Demographic characteristics of each study group.**

|  | DM, n = 16 | NDM, n = 15 | p value |
|---|---|---|---|
| Age, years | 66.6 ± 10.6 | 69.4 ± 10.7 | 0.49 |
| Enrolled eyes | 30 | 30 |  |
| Sex, (%) |  |  |  |
| Male | 9 (57) | 7 (47) |  |
| Female | 7 (43) | 8 (53) |  |
| Spherical equivalent, D | 0.15±1.9 | 0.76±1.77 | 0.30 |
| BCVA, logMAR | 0.17±0.18 | 0.08±0.14 | 0.046 |
| Axial length, mm | 23.50 ± 0.96 | 23.18 ± 1.12 | 0.33 |
| Body weight change, kg | 3.63 ± 1.59 | 2.78 ± 2.20 | 0.28 |
| DR severity, eyes (NDR/NPDR/PDR) | 8/16/6 |  |  |
| PRP treatment, eyes | 6 |  |  |

Hypertensive nephrosclerosis = 8; IgA nephropathy = 2; Multiple myeloma = 1; Unknown etiology = 4.

**Abbreviations**: DM, diabetes mellitus; NDM, nonDM; BCVA, best-corrected visual acuity; DR, diabetic retinopathy; NDR, no diabetic retinopathy; NPDR, nonproliferative DR; PDR, proliferative DR; PRP, pan-retinal photocoagulation.

**Table 2. Changes in ocular parameters before and after the initiation of hemodialysis.**

| | DM group, n = 30 | | NDM group n = 30 | |
|---|---|---|---|---|
| | Before hemodialysis | After hemodialysis | Before hemodialysis | After hemodialysis |
| IOP, mmHg | 12.0 ± 3.0 | 11.2 ± 2.7† | 11.3 ± 1.7 | 10.7 ± 1.4* |
| Ocular perfusion pressure, mmHg | 60.3 ± 11.5 | 59.4 ± 7.8 | 61.2 ± 7.9 | 51.7 ± 4.8† |
| Spherical equivalent, D | 0.15 ± 1.9 | 0.09 ± 1.8 | 0.76 ± 1.8 | 0.76 ± 1.8 |
| AL, mm | 23.5 ± 0.96 | 23.6 ± 0.88 | 23.2 ± 1.1 | 23.2 ± 1.1 |
| ACD, mm | 3.13 ± 0.6 | 3.11 ± 0.6 | 3.13 ± 0.6 | 3.12 ± 0.6 |
| Central macular thickness, μm | 277.5 ± 51.7 | 276.4 ± 49.8 | 255.6 ± 17.4 | 255.3 ± 20.4 |

Before vs. after the initiation of hemodialysis (Wilcoxon signed-rank test);

*$p < 0.05$,

†$p < 0.01$.

**Abbreviations**: DM, diabetic; NDM, nondiabetic; IOP, intraocular pressure; AL, axial length; ACD, anterior chamber depth.

## Changes in ocular parameters before and after initiation of hemodialysis

Mean IOP decreased from 12.0 ± 3.0 mmHg to 11.2 ± 2.7 mmHg (p = 0.003) in the DM group. Mean ocular perfusion pressure decreased from 61.2 ± 7.9 mmHg to 51.7 ± 4.8 mmHg (p = 0.001) in the NDM group, whereas there were no changes in the DM group. Mean central macular thickness was 277 ± 51.7 μm and 276 ± 49 μm in the DM and NDM groups, respectively, prior to hemodialysis initiation. These values decreased to 255 ± 17 μm and 255 ± 20 μm in the DM and NDM groups, respectively, following the completion of 2 weeks of dialysis. There was no significant difference between these values. Similarly, no significant between-group differences were observed in spherical equivalent, AL, or ACD (Table 2).

## Effect of initiation of hemodialysis on choroidal thickness and layer analysis

Mean SCT and mean LCVLT decreased in both groups after hemodialysis initiation. In the DM group, mean SCT before and after hemodialysis initiation decreased from 301.7 ± 70.3 μm to 261.6 ± 77.3 μm, and mean LCVLT decreased from 256.8 ± 66.6 μm to 217.7 ± 72.6 μm (Table 3). Mean SCT decreased from 281.8 ± 57.1 μm to 259.8 ± 68.3 μm and

**Table 3. Changes in each layer and area of the choroid before and after the initiation of hemodialysis.**

| | DM | | | NDM | | |
|---|---|---|---|---|---|---|
| | Before hemodialysis | After hemodialysis | p | Before hemodialysis | After hemodialysis | p |
| **Choroidal thickness, μm** | | | | | | |
| Total | 301.7 ± 70.3 | 261.6 ± 77.3 | <0.001 | 281.8 ± 57.1 | 259.8 ± 68.3 | <0.001 |
| CC+MCVLT | 45.0 ± 10.2 | 43.7 ± 10.9 | 0.22 | 43.6 ± 10.8 | 43.1 ± 8.9 | 0.25 |
| LCVLT | 256.8 ± 66.6 | 217.7 ± 72.6 | <0.001 | 238.8 ± 56.4 | 216.7 ± 66.0 | <0.001 |
| **Choroidal area, mm²** | | | | | | |
| Total | 1.75 ± 0.23 | 1.36 ± 0.18 | <0.001 | 1.51 ± 0.15 | 1.25 ± 0.17 | <0.001 |
| Luminal | 1.30 ± 0.18 | 0.94 ± 0.13 | <0.001 | 1.18 ± 0.11 | 0.97 ± 0.10 | <0.001 |
| Stromal | 0.45 ± 0.15 | 0.41 ± 0.16 | 0.058 | 0.32 ± 0.14 | 0.28 ± 0.08 | 0.065 |

Before vs. after the initiation of hemodialysis (generalized linear mixed model).

**Abbreviations**: DM, diabetic; NDM, nondiabetic; CC+MCVLT, choriocapillaris-medium choroidal vessel layer thickness; LCVLT, large choroidal vessel layer thickness.

**Table 4. Change ratio of each layer and area of the choroid before and after the initiation of hemodialysis.**

|  | DM | NDM | p |
|---|---|---|---|
| ΔChoroidal thickness, Mm |  |  |  |
| SCT | 40.1 ± 20.8 (13.3%) | 22.0 ± 18.5 (9.5%) | 0.049 |
| CC+MCVLT | 1.4 ± 0.9 (3.1%) | 1.9 ± 1.1 (4.1%) | 0.12 |
| LCVLT | 38.5 ± 19.8 (14.5%) | 21.9 ± 17.9 (9.2%) | 0.02 |
| ΔChoroidal area, mm$^2$ |  |  |  |
| Total | 0.38 ± 0.23 (21.9%) | 0.26 ± 0.17 (17.2%) | 0.032 |
| Luminal | 0.36 ± 0.15 (27.7%) | 0.21 ± 0.12 (17.7%) | 0.007 |
| Stromal | 0.04 ± 0.02 (10.0%) | 0.04 ± 0.03 (12.3%) | 0.055 |

ΔChoroidal thickness (area): (thickness before HD—thickness after HD) of each layer (area) of the subfoveal choroid. DM vs. NDM (generalized linear mixed model).

**Abbreviations**: DM, diabetic; NDM, nondiabetic; CC+MCVLT, choriocapillaris-medium choroidal vessel layer thickness; LCVLT, large choroidal vessel layer thickness; SCT, subfoveal choroidal thickness.

mean LCVLT decreased from 238.8 ± 56.4 μm to 216.7 ± 66.0 μm in the NDM group (Table 3). No significant between-group differences were observed in the mean CC + MCVLT (Table 3). Mean SCT values in the DM and NDM groups were −13.3% ± 2.5% and −9.5% ± 3.1%, respectively, with a significant decrease in the DM group (Table 4). Mean LCVLT values in the DM and NDM groups was −14.5% ± 5.2% and −9.2% ± 3.4%, respectively, with a significant decrease in the DM group (Table 4).

## Effect of initiation of hemodialysis on choroidal area, luminal area, and stromal area

After initiation of hemodialysis, mean SCA and mean LA decreased in both groups (Table 3). In the DM group, mean SCA before and after the initiation of hemodialysis decreased from 1.75 ± 0.23 mm$^2$ to 1.36 ± 0.18 mm$^2$ and the mean LA decreased from 1.30 ± 0.18 mm$^2$ to 0.94 ± 0.13 mm$^2$ (Table 3). In the NDM group, mean SCA decreased from 1.51 ± 0.15 mm$^2$ to 1.25 ± 0.17 mm$^2$ and mean LA decreased from 1.18 ± 0.11 mm$^2$ to 0.97 ± 0.10 mm$^2$ (Table 3). There was no significant change in mean SA in either group (Table 3). The mean SCA was −21.9% ± 6.5% and −17.2% ± 5.9% in the DM and NDM groups, respectively, with a significant decrease in the DM group (Table 4). Mean LA was −27.7% ± 8.7% and −17.7% ± 5.8% in the DM and NDM groups, respectively, with a significant decrease in the DM group (Table 4).

## Discussion

This is the first study to investigate structural changes in the choroid of patients receiving hemodialysis for the first time. Additionally, this is the first evaluation that demonstrated the occurrence of significant changes in the SCT, LCVLT, SCA, and LA in patients with diabetes.

Hemodialysis influences the choroid, as shown by numerous studies. Ishibazawa et al. described that hemodialysis facilitates body fluid removal and decreases choroidal thickness, measured using SD-OCT without EDI [19]. Chang et al., using EDI-OCT, observed that reductions in choroidal thickness correlate with reductions in body weight, systolic blood pressure, and serum osmolarity [21]. The reports by Ishizawa et al. and Chang et al. demonstrated that mean choroidal thickness changes are greater in patients with diabetes. Shin et al., using SS-OCTA, demonstrated reductions in total perfused vessel density in the choriocapillaris and total choroidal thickness following hemodialysis [22]. Although Schocket et al. reported that

choroidal thickness and choroidal blood flow decrease in PDR eyes, [23] other studies report that such reductions are unrelated to the severity of DR [24]. Therefore, this relationship remains controversial, and in the present study, we primarily evaluated changes in the choroid.

SCT decreased in both the DM and NDM groups at 2 weeks after the initiation of hemodialysis, whereas LCVLT decreased in both groups. Intra- and extravascular fluid retention in patients with ESKD may cause decreases in plasma and stromal volume owing to increased colloid osmotic pressure and formation of a transluminal osmotic pressure gradient [4]. This may cause a decrease in choroidal thickness owing to fluid removal from blood vessels and the choroidal stroma after a single session of hemodialysis [19]. This mechanism may also be attributable for similar findings in patients undergoing hemodialysis for the first time.

Previous studies noted that pathological vascular disorders may cause leakage of large molecules such as albumin, exacerbating changes in the choroidal thickness in the LCVL. Electron microscopic observations have revealed endothelial epilamellar disorders in deeper areas and leakage of proteinaceous fluid into the choroidal stroma in patients with diabetic choroidopathy [25].

In the present study, the choroid was analyzed after identification of vascular plexus and interstitial tissue by using the binarization method [17, 18]. Preeti et al. reported significant alterations in choroidal structure and vascular characteristics in patients with DR. They believed that this was due to reduced vascularity, which is consistent with capillary dropout, a process in which reduced choroidal blood flow plays a key role in pathogenesis [26].

We observed large changes in both the SCA and LA in the DM group. These changes, as described in previous reports [7, 8], were likely due to significant changes in blood vessels caused by worsening diabetic choroidopathy against a backdrop of reduction in choroidal blood following the onset of narrowing of the choroidal arterioles, choriocapillaris atrophy, and capillary dropout of the choriocapillaris. Additionally, the high permeability of the choroid might also contribute to these changes. The choriocapillaris beneath Bruch's membrane is well-fenestrated [27], which allows small molecules such as glucose and amino acids to leak out. Therefore, the increased permeability of the choroid may be significant both before and after initiation of hemodialysis in the DM group from the effects of diabetic choroidopathy.

Our study had several limitations. First, our sample size was small, and further studies with larger sample sizes are needed. Second, we compared data from before the first and after the last hemodialysis sessions, which were separated by approximately 2 weeks, to evaluate changes in patients who had undergone initiation of dialysis for the first time. A study comparing data from a longer interval, such as 1–3 months, might reveal greater changes in the choroid. Third, patients who had undergone PRP treatment were also included in the DM group. A previous study [28] found that retinal photocoagulation directly affects the choriocapillaris. However, changes to choroidal circulation following photocoagulation remain controversial [29], since a study using scanning electron microscopy found no damage in large vessels in the deeper choroid following retinal photocoagulation [30]. We found no significant changes in the CC+MCVLT before and after hemodialysis in both groups. The effects of PRP may be limited to total choroidal thickness, total choroidal area, LCVLT, and the measurable LA.

In future studies, it may be necessary to obtain additional evidence related to choroidal compensatory function in each stage of DR.

In summary, SCT and SCA significantly decreased in both groups following hemodialysis, and decreases in LCVLT and the LA were large following stratification and binarization. The changes in SCT and LCVLT and the changes in SCA and LA were larger in the DM group than in the NDM group.

## Supporting information

**S1 Fig. Schedule of patients who were newly introduced into hemodialysis.** Approximately 6 rounds of hemodialysis were performed during a ~2-week hospital stay. Before the initiation of HD: measurement obtained before the first hemodialysis. After initiation of HD: measurement obtained after the last hemodialysis of the hospital stay.
(TIF)

**S1 Table. Changes in systemic parameters before and after the initiation of hemodialysis.**
(DOCX)

## Acknowledgments

The authors would like to acknowledge true team assistance from the Niigata City General Hospital and Uonuma Kikan Hospital. We would like to thank Editage (www.editage.com) for English language editing.

## Author Contributions

**Conceptualization:** Hideyuki Nakano, Hiruma Hasebe, Takeo Fukuchi.

**Data curation:** Hideyuki Nakano, Hiroyuki Cho, Daisuke Kondo, Noriaki Iino.

**Formal analysis:** Hideyuki Nakano.

**Investigation:** Hideyuki Nakano, Kenji Murakami, Hiroyuki Cho, Daisuke Kondo, Noriaki Iino.

**Methodology:** Hideyuki Nakano, Hiruma Hasebe, Daisuke Kondo, Noriaki Iino.

**Project administration:** Hiruma Hasebe, Takeo Fukuchi.

**Resources:** Kenji Murakami, Hiroyuki Cho, Daisuke Kondo, Noriaki Iino.

**Supervision:** Hiruma Hasebe, Takeo Fukuchi.

**Validation:** Hideyuki Nakano, Hiruma Hasebe.

**Writing – original draft:** Hideyuki Nakano, Hiruma Hasebe, Takeo Fukuchi.

**Writing – review & editing:** Hideyuki Nakano, Hiruma Hasebe, Takeo Fukuchi.

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
