## [Decision Letter · Decision Letter 0]

10 Jul 2020

PONE-D-20-09258

Choroid structure analysis following initiation of hemodialysis by using swept-source optical coherence tomography in patients with and without diabetes

PLOS ONE

Dear Dr. Nakano,

Thank you for submitting your manuscript to PLOS ONE. After careful consideration, we feel that it has merit but does not fully meet PLOS ONE’s publication criteria as it currently stands. Therefore, we invite you to submit a revised version of the manuscript that addresses the points raised during the review process.

A learned reviewer have offered a few minor criticisms that can be easily addressed by incorporating appropriate changes in the manuscript. 

We look forward to receiving your revised manuscript.

Kind regards,

Sanjoy Bhattacharya

Academic Editor

PLOS ONE

Journal Requirements:

Reviewers' comments:

Reviewer's Responses to Questions

**Comments to the Author**

1. Is the manuscript technically sound, and do the data support the conclusions?

Reviewer #1: Yes

2. Has the statistical analysis been performed appropriately and rigorously? 

Reviewer #1: Yes

3. Have the authors made all data underlying the findings in their manuscript fully available?

Reviewer #1: Yes

4. Is the manuscript presented in an intelligible fashion and written in standard English?

Reviewer #1: Yes

5. Review Comments to the Author

Reviewer #1: Nakano et al present a multi-centre prospective study evaluating choroidal structural changes using SS-OCT following the initiation of hemodialysis in patients with and without diabetes. Overall, the study appears to be well-written, and addresses the research question appropriately. The Methods section describes the SS-OCT imaging and interpretation procedure appropriately. It appears that the choroidal thickness and area are significantly decreased in all patients following hemodialysis, with this effect pronounced in those with diabetes. I believe this manuscript is suitable for publication following some minor revisions.

Specific comments:

1. Methods – Line 92 – this statement may imply that patients underwent hemodialysis prior to SS-OCT, please revise to clarify that all patients were assessed prior to their first dialysis session.

2. Results – the authors do a good job of delineating the changes in systemic parameters following hemodialysis in both groups. It may be worthwhile to use a MANOVA to determine if there is a between groups effect (ie. To see if the effect of dialysis was different in DM versus non-DM groups). (Supp Table 1)

3. Table 1 – consider including visual acuity, ocular comorbidities and systemic comorbidities in this Table

4. Table formatting – in some tables, the numerical values are separated across lines. This is a minor typesetting issue only, but affects readability.

6. PLOS authors have the option to publish the peer review history of their article (what does this mean?). If published, this will include your full peer review and any attached files.

Reviewer #1: **Yes: **Vinay Kansal

---

## [Author Response · Author response to Decision Letter 0]

17 Aug 2020

Response to Reviewers

Specific comments:

1. Methods – Line 92 – this statement may imply that patients underwent hemodialysis prior to SS-OCT, please revise to clarify that all patients were assessed prior to their first dialysis session.

Response: Thank you for your helpful suggestion. We have now included this information in the “Study subjects” subsection of the manuscript.

2. Results – the authors do a good job of delineating the changes in systemic parameters following hemodialysis in both groups. It may be worthwhile to use a MANOVA to determine if there is a between groups effect (ie. To see if the effect of dialysis was different in DM versus non-DM groups). (Supp Table 1)

Response: Thank you for your comment. In the present study, we primarily compared the intra-group ocular parameters. We plan to leave the inter-group systemic parameter comparison as an objective for additional studies in the future. As you have suggested, MANOVA would be a useful technique for such inter-group comparisons in future studies.

3. Table 1 – consider including visual acuity, ocular comorbidities and systemic comorbidities in this Table

Response: With regard to visual acuity (logMAR), we have added the relevant data to Table 1—DM group: 0.17±0.18 and NDM group: 0.08±0.14 (p=0.046). Regarding ocular comorbidities, we have mentioned in the “Study Subjects” subsection that patients with anterior or posterior segment disease that hindered accurate examinations, those who had undergone ocular surgery and/or retinal laser treatments within 3 months prior to the study, and those who had a history of retinal vein occlusion, glaucoma, AMD, or uveitis were excluded from this study. Data regarding DR severity and number of PRP treatments have also been provided in Table 1. Underlying diseases related to renal failure in the NDM group have been listed as systemic comorbidities in the Table 1 footnote. Please refer to the corresponding sections in the revised manuscript for these data.

4. Table formatting – in some tables, the numerical values are separated across lines. This is a minor typesetting issue only, but affects readability.

Response: Thank you for noting this issue; the required revisions has been made accordingly.

---

## [Decision Letter · Decision Letter 1]

31 Aug 2020

Choroid structure analysis following initiation of hemodialysis by using swept-source optical coherence tomography in patients with and without diabetes

PONE-D-20-09258R1

Dear Dr. Nakano,

We’re pleased to inform you that your manuscript has been judged scientifically suitable for publication and will be formally accepted for publication once it meets all outstanding technical requirements.

Kind regards,

Sanjoy Bhattacharya

Academic Editor

PLOS ONE

Additional Editor Comments (optional):

Reviewers' comments:

Reviewer's Responses to Questions

**Comments to the Author**

1. If the authors have adequately addressed your comments raised in a previous round of review and you feel that this manuscript is now acceptable for publication, you may indicate that here to bypass the “Comments to the Author” section, enter your conflict of interest statement in the “Confidential to Editor” section, and submit your "Accept" recommendation.

Reviewer #1: All comments have been addressed

2. Is the manuscript technically sound, and do the data support the conclusions?

Reviewer #1: Yes

3. Has the statistical analysis been performed appropriately and rigorously? 

Reviewer #1: Yes

4. Have the authors made all data underlying the findings in their manuscript fully available?

Reviewer #1: Yes

5. Is the manuscript presented in an intelligible fashion and written in standard English?

Reviewer #1: Yes

6. Review Comments to the Author

Reviewer #1: Nakano et al present a multi-centre prospective study evaluating choroidal structural changes using SS-OCT following the initiation of hemodialysis in patients with and without diabetes. Overall, the study appears to be well-written, and addresses the research question appropriately. The Methods section describes the SS-OCT imaging and interpretation procedure appropriately. It appears that the choroidal thickness and area are significantly decreased in all patients following hemodialysis, with this effect pronounced in those with diabetes

7. PLOS authors have the option to publish the peer review history of their article (what does this mean?). If published, this will include your full peer review and any attached files.

Reviewer #1: **Yes: **Vinay Kansal

---

## [Editor Report · Acceptance letter]

4 Sep 2020

PONE-D-20-09258R1 

Choroid structure analysis following initiation of hemodialysis by using swept-source optical coherence tomography in patients with and without diabetes 

Dear Dr. Nakano:

I'm pleased to inform you that your manuscript has been deemed suitable for publication in PLOS ONE. Congratulations! Your manuscript is now with our production department. 

Kind regards, 

on behalf of

Dr. Sanjoy Bhattacharya 

Academic Editor

PLOS ONE